**High-Resolution Meteorological Forcing Data for Hydrological Modelling and Climate Change**
**Impact Analysis in Mackenzie River Basin**
**Zilefac Elvis Asong[1], Mohamed Ezzat Elshamy[1], Daniel Princz[1], Howard Simon Wheater[1], John W.**
**Pomeroy[1, 2], Alain Pietroniro[1, 2, 3] and Alex Cannon[4]**
[1]*Global Institute for Water Security, University of Saskatchewan, 11 Innovation Blvd, Saskatoon, SK,*
*Canada S7N 3H5*
[2]*Centre for Hydrology, University of Saskatchewan, 121 Research Drive, Saskatoon, SK, Canada S7N 1K2*
[3]*Environment and Climate Change Canada, 11 Innovation Blvd, Saskatoon, SK, Canada S7N 3H5*
[4]*Climate Research Division, Environment and Climate Change Canada, BC V8W 2Y2, Victoria, Canada*
*****Corresponding author:**
Phone: +1 306 491 9565
Email: *elvis.asong@usask.ca*
**Abstract**
Cold regions hydrology is very sensitive to the impacts of climate warming. Impacts of warming over
recent decades in western Canada include glacier retreat, permafrost thaw and changing patterns of
precipitation, with an increased proportion of winter precipitation falling as rainfall and shorter durations
of snowcover, and consequent changes in flow regimes. Future warming is expected to continue along
these lines. Physically realistic and sophisticated hydrological models driven by reliable climate forcing can
provide the capability to assess hydrological responses to climate change. However, the provision of
reliable forcing data remains problematic, particularly in data sparse regions. Hydrological processes in
cold regions involve complex phase changes and so are very sensitive to small biases in the driving
meteorology, particularly in temperature and precipitation, including precipitation phase. Cold regions
often have sparse surface observations, particularly at high elevations that generate a large amount of
runoff. This paper aims to provide an improved set of forcing data for large scale hydrological models for
climate change impact assessment. The best available gridded data in Canada is from the high-resolution
forecasts of the Global Environmental Multiscale (GEM) atmospheric model and outputs of the Canadian
Precipitation Analysis (CaPA) but these datasets have a short historical record. The EU WATCH ERA-Interim
reanalysis (WFDEI) has a longer historical record but has often been found to be biased relative to
observations over Canada. The aim of this study, therefore, is to blend the strengths of both datasets
(GEM-CaPA and WFDEI) to produce a less-biased long record product (WFDEI-GEM-CaPA) for hydrological
modelling and climate change impacts assessment over the Mackenzie River Basin. First, a multivariate
generalization of the quantile mapping technique was implemented to bias-correct WFDEI against GEM-
CaPA at 3h × 0.125° resolution during the 2005-2016 overlap period, followed by a hindcast of WFDEI-
GEM-CaPA from 1979. The derived WFDEI-GEM-CaPA data are validated against station observations as a
preliminary step to assess their added value. This product is then used to bias-correct climate projections
from the Canadian Centre for Climate Modelling and Analysis Canadian Regional Climate Model
(CanRCM4) between 1950 – 2100 under RCP8.5, and an analysis of the datasets shows that the biases in
the original WFDEI product have been removed and the climate change signals in CanRCM4 are preserved.
The resulting bias-corrected datasets are a consistent set of historical and climate projection data suitable
for large-scale modelling and future climate scenario analysis. The final historical product (WFDEI-GEM-
CaPA,   1979-2016)   is   freely   available   at   the   Federated   Research   Data   Repository   at
http://dx.doi.org/10.20383/101.0111 (Asong et al., 2018) while the original and corrected CanRCM4 data
are available at https://doi.org/10.20383/101.0162 (Asong et al., 2019).
**Subject Keywords:** cold regions processes, observations, bias correction, Mackenzie River Basin














## 1    Introduction

**1    Introduction**
Accurate and reliable weather and climate information at the basin scale is in increasingly high
demand by policy-makers, scientists, and other stakeholders for many purposes including water resources
management (Barnett et al., 2005), infrastructure planning  (Brody et al., 2007), and ecosystem modelling
(IPCC, 2013). Specifically, the potential impacts of a warming climate on water availability in snow-
dominated high latitude regions continue to be a serious concern given that over the past several decades,
these regions have experienced some of the most rapid warming on earth (Demaria et al., 2016;
Diffenbaugh et al., 2012; Islam et al., 2017; Martin and Etchevers, 2005; Stocker et al., 2013). The on-going
science suggests that these warming trends are resulting in the intensification of the hydrologic cycle,
leading to significant recent observed changes in the hydro-climatic regimes of major river basins in
Canada and globally (Coopersmith et al., 2014; DeBeer et al., 2016; Dumanski et al., 2015). Changes in the
timing and magnitude of river discharge (Dibike et al., 2016), shifts in extreme temperature and
precipitation regimes (Asong et al., 2016b; Vincent et al., 2015) and changes in snow, ice, and permafrost
regimes are anticipated (IPCC, 2013). Substantial evidence also indicates that the long-held notion of
stationarity of hydrological processes is becoming invalid in a changing climate.  As pointed out by Milly
et al. (2008), this loss of stationarity means that there will be an increase in the likelihood and frequency
of extreme weather and climate events, including floods and droughts.   What is particularly troubling is
that these impacted regions typically have extremely low density of weather and climate observations,
making any attribution and climate impact analysis on water resources difficult.
It is well understood that water resources in most watersheds north of 30° N are heavily
dependent on natural water storage provided by snowpacks and glaciers, with water accumulated in the
solid phase during the cold season and released in the liquid phase during warm events and the warm
season. Particularly, the Canadian Rocky Mountains, the hydrological apex of North America with
headwater streams flowing to the Arctic, Atlantic and Pacific oceans, constitutes an integral part of the
global hydrological cycle (Fang et al., 2013). Flows in these high elevation headwaters depend heavily on
meltwater from snowpacks and glaciers. However, given that it is characterized by a highly varying cold
region hydroclimate, studies indicate that it is in these high elevation regions where climate variability
and change is expected to be most pronounced in terms of its impacts on water supply (Beniston, 2003;
Kane et al., 1991; Prowse and Beltaos, 2002; Woo and Pomeroy, 2011). More physically realistic and
sophisticated hydrological models driven by reliable climate forcing information can enhance our ability
to assess short- and long-term regional hydrologic responses to increasing variability and uncertainty in
hydro-climatic conditions in a changing climate. Nonetheless, hydrological processes in cold regions
involve complex phase changes and so are very sensitive to small biases in the driving meteorology,
particularly in temperature and precipitation.

As described earlier, cold regions often have sparse surface observations, particularly at the high

elevations and high latitudes regions that could potentially generate a major amount of runoff. The effects
of mountain topography and high latitudes are currently not well reflected in the observational record.
Ground-based measurements (e.g. gauges) are limited especially over the Canadian Rocky Mountains,
and suffer from inaccuracies associated with cold climate processes (Asong et al., 2017a; Wang and Lin,
2015; Wong et al., 2017). The advent and use of weather radar systems have addressed some of the
shortcomings of gauge coverage, at least where radar exists. Unfortunately, in Canada, for example, the
spatial coverage of weather radar is limited to the southern (south of 55° N) part of the country (Fortin et
al., 2015b). Recently, improved satellite products have emerged such as the Global Precipitation
Measurement (GPM) mission that provides meteorological information at fine spatiotemporal resolutions
and regular intervals. But, the GPM is still at an early stage of development and only covers the region
south of 60° N (Asong et al., 2017a; Hou et al., 2014).

The capability of the current generation of Earth System Models (ESMs) to represent

meteorological forcing variables is therefore of major interest for hydrological climate change impact
studies in cold regions watersheds. Despite commendable progress being made, raw outputs from
regional and global ESMs still have large differences between models and from the limited observational
reference meteorology, due partly to spatial scale mismatches and systematic biases (Taylor et al., 2012).
Therefore, ESM outputs are often downscaled and biases are adjusted statistically before being used in
hydrological simulations (Asong et al., 2016b; Chen et al., 2013; Chen et al., 2018; Gudmundsson et al.,
2012). Recent research has demonstrated that bias correction, including adjustment of the dependence
between driving variables, can lead to more realistic hydrological simulations in cold regions watersheds
where the response of the system is sensitive to accumulation and melt of snow and ice (Meyer et al.,

2019).

Apart from the uncertainty due to the many empirical statistical techniques which have been

developed to post-process ESM outputs (Maraun, 2016), the quality and length of the reference
observational dataset for bias correction remains a major issue (Reiter et al., 2016; Schoetter et al., 2012;
Sippel et al., 2016). In Canada and other regions of North America, regional gridded datasets such as the
combined Global Environmental Multiscale (GEM) atmospheric model forecasts (Yeh et al., 2002) and the
Canadian Precipitation Analysis—CaPA (Mahfouf et al., 2007)  have been found to perform comparably to
ground observations, both statistically and hydrologically (Alavi et al., 2016; Boluwade et al., 2018; Eum
et al., 2014; Fortin et al., 2015a; Gbambie et al., 2017; Wong et al., 2017). However, the duration of GEM-
CaPA is too short to be used to directly correct ESM climate due to unsynchronized internal
variability—the recommended minimum record length for bias correction is 30 years (Maraun, 2016;
Maraun et al., 2017). Other gridded products such as the EU WATCH ERA-Interim reanalysis—WFDEI
(Weedon et al., 2014) and Princeton (Sheffield et al., 2006) have a longer historical record, but have been
found to be biased relative to observations over Canada (Wong et al., 2017) and the United States (Behnke
et al., 2016; Sapiano and Arkin, 2009). However, the WFDEI reanalysis has been found to outperform other
long-record gridded products (Chadburn et al., 2015; Park et al., 2016; Wong et al., 2017).
Because of the sparse observational network, few gridded climate datasets exist that contain the
necessary meteorological variables to drive physically-based land surface models at sub-daily temporal
resolution north of 55° N in North America. Because the combination of the GEM and CaPA datasets has
been shown to perform relatively well in these regions, the intent here is to use these datasets to bias-
correct the WFDEI dataset, which contains a sufficient length of record for bias-correcting climate
projection datasets. Aside from its short record length, a limitation of the GEM-CaPA dataset for wider
use for hydrological models is that the wind, temperature, and humidity variables are available only at
the 0.995 sigma($\sigma$) level (approximately 40 m, varying in time and space; herein referred to as the "40 m"
level) across the full length of record. The WFDEI dataset contains these variables at the surface level,
which is more typically used by hydrological models. Therefore, the bias correction effectively modifies
the source surface level data to reproduce the climate found at the 40 m level of the reference dataset
(GEM-CaPA). Many regional and large-scale land surface hydrological models are capable of using climate
data at this atmospheric level. Thus, no effort is made to interpolate the product back to surface level
(although this could be done if needed). In addition, the bias-corrected dataset at an effective 40 m level
can then be used to bias-correct these same fields from the CanRCM4 dataset, which are at the same
0.995 $\sigma$ level as in the reference dataset (GEM-CaPA). The analysis results in a bias-corrected set of
historical and projected climate data that is consistent in time and considers the regional topography and
climate effects of GEM and CaPA, and is suitable to drive large-scale simulations of distributed hydrological
models for assessing climate change impacts in data sparse regions.
The aim of this study, therefore, is to combine the strengths of both datasets (GEM-CaPA and
WFDEI) to produce a less-biased long record product (WFDEI-GEM-CaPA) using a multi-stage bias
correction framework. First, a multivariate generalization of the quantile mapping technique was
implemented to bias-correct WFDEI against GEM-CaPA at 3h × 0.125° resolution during the 2005-2016
period, followed by a hindcast of WFDEI-GEM-CaPA from 1979. Subsequently, a 15-member initial
condition ensemble of the CanESM2 ESM (historical followed by RCP8.5 scenario), which has been
dynamically downscaled at 0.44° (50 km) resolution using the fourth generation Canadian Regional
Climate Model (CanRCM4), is sourced from the Canadian Centre for Climate Modelling and Analysis. A
multivariate bias correction algorithm is applied to the CanRCM4 outputs (1950 – 2100) to adjust the data
against WFDEI-GEM-CaPA. The bias-corrected products are important for developing distributed
hydrological models as well as for assessing climate change impacts over the Mackenzie River basin (MRB),
which constitutes a testbed for the Changing Cold regions Network (CCRN) project's large-scale
hydrological modelling strategy and is the case study for the current analysis.
**2      Methodology**
**2.1      Study area**
The study area is the Mackenzie River Basin (MRB), which is the largest river basin in Canada and the
largest river draining from North America to the Arctic Ocean (
**Fig. 1**). It drains an area of about 1.8 million $km^2$ and discharges more than 300 $km^3$ of freshwater

to the Beaufort Sea in the Arctic each year. The basin drains parts of British Columbia, Alberta,
Saskatchewan, the Northwest Territories and the Yukon Territory in northwestern Canada.  The western
tributaries are relatively steep as they originate from the Canadian Rocky Mountains while the eastern
tributaries have milder topography with several large lakes, thousands of interconnected small lakes, fens
and bogs. The general vegetation ranges widely between alpine, boreal and tundra landscapes. Climatic
conditions are also quite variable and can be generally classified as cold-temperate, mountain, subarctic
and arctic zones, with about 75% of the basin underlain by continuous and discontinuous permafrost.

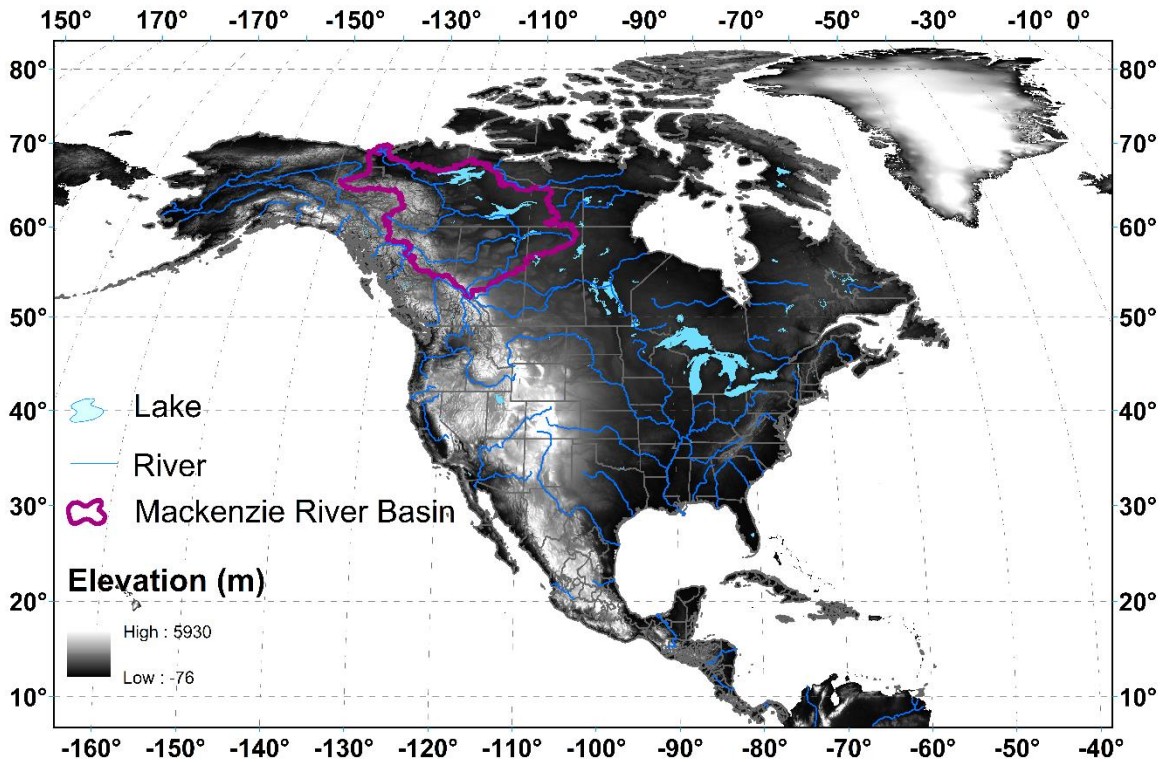

**Fig. 1**: Location of the Mackenzie River Basin in North America

**2.2      Data sources**
**2.2.1    Gridded GEM-CaPA product**

Hourly archived forecast data from the GEM model were acquired from Environment and Climate

Change   Canada   (http://collaboration.cmc.ec.gc.ca/cmc/cmoi/product_guide/submenus/rdps_e.html,
last access: 28 September 2018). The fields include downward incoming solar radiation, downward
incoming longwave radiation and pressure at the surface, as well as specific humidity, air temperature,
and wind speed at approximately 40 m above ground surface. The 40 m level was used because surface
level variables at 1.0 σ (approximately at 2 m for temperature and humidity, and 10 m for wind speed)
are only available in the archive from 2010 onward. The GEM data are at approximately 24 km spatial
resolution from October 2001, approximately 15 km from June 2004, and approximately 10 km resolution
from November 2012, and are provided on a rotated latitude/longitude grid in Environment and Climate
Change Canada (ECCC) 'standard file' format. The archived data are of former operational forecasts and
contain model outputs from versions of GEM prior to 2.0.0 through 5.0.0.
6-Hourly total precipitation data from the complementary CaPA product
(http://collaboration.cmc.ec.gc.ca/cmc/cmoi/product_guide/submenus/capa_e.html, last access: 28
September 2018) were also acquired. The analysis incorporates observed precipitation from
meteorological weather stations, and more recently from radar, into the precipitation field from GEM.
The CaPA data are approximately 10-km resolution from January 2002, also on a rotated
latitude/longitude grid in ECCC 'standard file' format. The data contain reanalysis outputs from CaPA
2.4b8 from 2002-2012, and of former operational analyses from versions of CaPA 2.3.0 through 4.0.0 from
November 2012 onward.
The variables from GEM and CaPA were spatially interpolated and re-projected to a regular
latitude/longitude grid at 0.125° resolution. For data from GEM, the interpolation was done using a
bilinear algorithm, while data from CaPA were interpolated using nearest neighbor (Schulzweida et al.,
2004). Where necessary, the GEM fields were converted to SI units and CaPA was converted to a
precipitation rate in SI units for better compatibility with some hydrological models.
**2.2.2   Gridded WFDEI product**
The gridded WFDEI meteorological forcing data has a global 0.5° spatial resolution and 3-h time
step covering the period 1979-2016 (http://www.eu-watch.org/data_availability, last access: 25 July
2018). Weedon et al. (2014) used the ERA-Interim surface meteorology data as baseline information to
derive the WFDEI product. Firstly, ERA-Interim data were interpolated at half-degree spatial resolution to
match the land–sea mask defined by the Climatic Research Unit (CRU) of the University of East Anglia,
Norwich, England. Subsequently, corrections for elevation and monthly bias of climate trends in the ERA-
Interim fields were applied to the interpolated data. The WFDEI data have two sets of precipitation data:
the Global Precipitation Climatology Centre product (GPCC) and CRU Time Series version 3.1 (CRU TS3.1).
Thus, two variants of the WFDEI product are available—WFDEI-GPCC and WFDEI-CRU. The WFDEI-CRU
dataset was used here because it goes up to 2016, whilst the WFDEI-GPCC had only been updated until
2013 at the time of our analysis.
**2.2.3     Station observations**
To evaluate the added value of bias-correcting WFDEI against GEM-CaPA, *in situ* hourly
precipitation, temperature, surface pressure, relative humidity, and wind speed at 773 stations located
across the MRB were initially considered (**Fig. 2**). This station network is maintained by Environment and
Climate Change Canada (ECCC) (http://climate.weather.gc.ca/historical_data/search_historic_
data_e.html, last access: 17 December 2019) and includes some duplicate stations (stations at the same
location but having different IDs). Total daily precipitation and average daily temperature are found in
daily data files while surface pressure, relative humidity, and wind speed are only found in hourly files.
Unfortunately, radiation data are not available at any of those stations. The data were extracted for the
period from 01 January 2005 to 31 December 2016 and hourly data were averaged to the daily time step.
This reduced the number of stations to 364. Out of these 364 stations, only 10 were found to have less
than 10% missing data (calculated at the daily timescale after aggregating/averaging the data) for all
studied variables concurrently over the 2005-2016 period and were retained for further consideration.
Precipitation and surface pressure are the only two surface variables in all datasets (gridded and stations).
Due to differences in heights between gridded variables of GEM-CaPA and WFDEI-GEM-CaPA datasets for
air temperature, humidity, and wind speed (see Sections 2.2.1 and 3.1) and the ECCC station data, we
expect deviations. Nevertheless, the comparisons are still informative. Relative humidity observations
were converted to specific humidity to be comparable to gridded datasets using concurrent station
temperature and surface pressure data at those stations, which reduced the record completeness further
but was still within 90% for the 10 selected stations. **Table 1** provides additional information for the 10
stations retained for further analysis, which are highlighted in **Fig. 2**. This dataset is hereafter referred to

as ECCC-S (S for stations). Table S1 in the Supplementary material provides a similar listing to **Table 1** but

for all 364 stations with records during the 2005-2016 period.

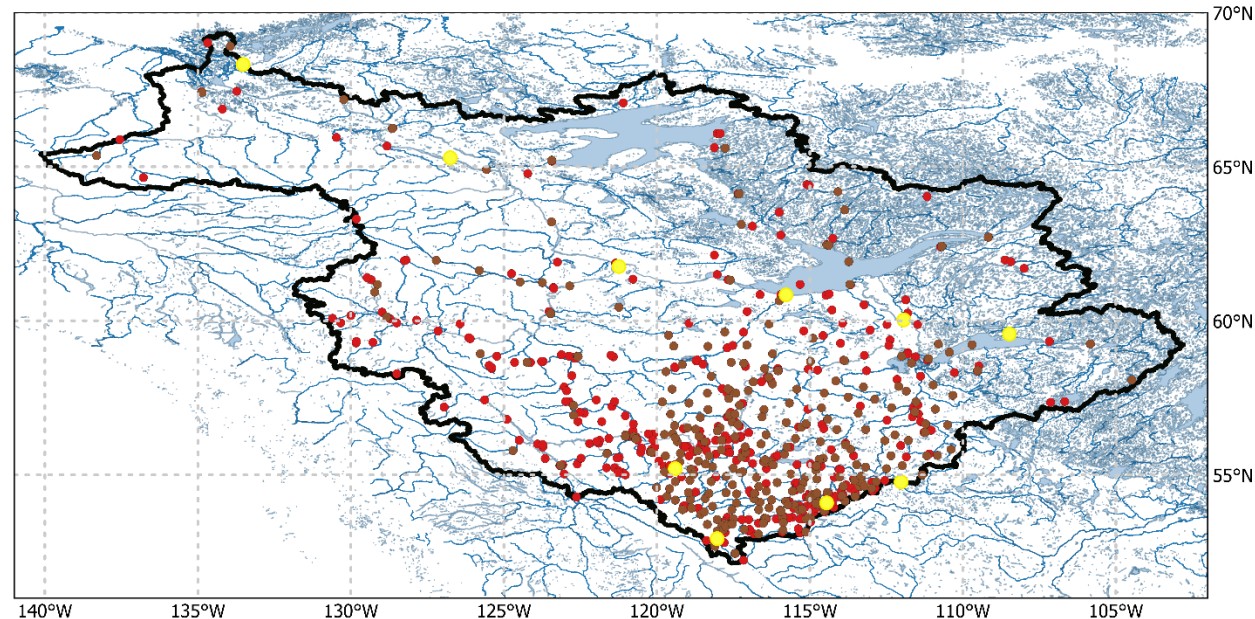

**Fig. 2**: Spatial distribution of the initial 773 ground-based precipitation gauges (all dots) over the study area. Only 364 of these have during the period 2005 – 2016 (brown & yellow dots). Data screening for missing values (10% threshold concurrently applied for all variables) during the 2005-2016 period resulted in 10 of these stations (yellow dots) being retained for validation of gridded datasets.

**Table 1**: List of observation stations used for validating the various gridded historical products

| | | Station | Coordinates | | | Record | | | | % Complete | | |
|---|---|---|---|---|---|---|---|---|---|---|---|---|
| Name | Prov. | ID | Lat | Long | Elev. | Start | End | T | P | RH | ps | wind |
| JASPER WARDEN | AB | 10223 | 52.93 | -118.03 | 1020.0 | 1994 | 2019 | 99.0 | 98.0 | 96.0 | 96.9 | 97.1 |
| BEAVERLODGE RCS | AB | 30669 | 55.20 | -119.40 | 745.0 | 2001 | 2019 | 99.0 | 93.9 | 93.8 | 93.9 | 93.8 |
| BARRHEAD CS | AB | 30641 | 54.09 | -114.45 | 648.0 | 2000 | 2019 | 98.2 | 98.1 | 97.8 | 97.8 | 97.0 |
| LAC LA BICHE CLIMATE | AB | 30726 | 54.77 | -112.02 | 567.0 | 2001 | 2019 | 99.0 | 98.8 | 97.8 | 97.9 | 97.9 |
| URANIUM CITY (AUT) | SK | 9831 | 59.57 | -108.48 | 318.2 | 1992 | 2019 | 95.8 | 93.0 | 94.2 | 94.4 | 94.8 |
| NORMAN WELLS CLIMATE | NT | 43004 | 65.29 | -126.75 | 93.6 | 2003 | 2019 | 98.5 | 96.6 | 96.0 | 95.4 | 96.2 |
| FORT SMITH CLIMATE | NT | 41884 | 60.03 | -111.93 | 203.0 | 2003 | 2019 | 97.6 | 96.8 | 95.8 | 96.7 | 97.3 |
| HAY RIVER CLIMATE | NT | 41885 | 60.84 | -115.78 | 164.0 | 2003 | 2019 | 99.6 | 99.3 | 98.4 | 98.4 | 98.0 |
| FORT SIMPSON CLIMATE | NT | 41944 | 61.76 | -121.24 | 168.0 | 2003 | 2019 | 97.5 | 99.5 | 96.1 | 96.2 | 98.2 |
| INUVIK CLIMATE | NT | 41883 | 68.32 | -133.52 | 103.0 | 2003 | 2019 | 99.6 | 95.1 | 98.3 | 98.4 | 97.0 |

**2.2.4 Climate model outputs**
The historical and future climate simulations utilized in this study are part of the CanRCM4 large
ensemble, which consists of 50 members and downscaled at horizontal spatial resolutions of 0.44° (~50
km). These CanRCM4 simulations had been produced initially by the Canadian Sea Ice and Snow Evolution
Network (CanSISE) Climate Change and Atmospheric Research (CCAR) Network project
(https://www.cansise.ca/, last access: 24 April 2019). The input data for the historical period, i.e., 1950 –
2005 as well as the future (2006 – 2100) RCP simulations of CanRCM4 were provided by the parent ESM
(CanESM2) as specified in the Coupled Model Intercomparison Project Phase 5 (CMIP5) guidelines. The
data are sourced from the Canadian Centre for Climate Modelling and Analysis (CCCma) at
www.cccma.ec.gc.ca/data/canrcm/CanRCM4 (last access: 6 March 2019). This study utilized 15 members
of the 0.44° resolution product at 1-h time step and values were aggregated to 3-h resolution prior to bias
correction. The seven forcing variables needed for driving the CCRN MESH model
(https://wiki.usask.ca/display/MESH/About+MESH, last access: 10 May 2019) and which were bias-
corrected in the current study are included in **Table 2**.
**Table 2**: List of variables processed in this study with heights and units in each dataset

| Variable | Unit | Height | | |
|---|---|---|---|---|
| | | WFDEI | GEM-CaPA | WFDEI-GEM-CaPA |
| Precipitation | kg m$^{-2}$ s$^{-1}$ | Surface | Surface | Surface |
| Air Temperature | K | 2 m | 40 m | 40 m |
| Specific Humidity | kg kg$^{-1}$ | 2 m | 40 m | 40 m |
| Wind Speed | m s$^{-1}$ | 10 m | 40 m | 40 m |
| Surface Pressure | Pa | Surface | Surface | Surface |
| Downwelling Shortwave Radiation | W m$^{-2}$ | Surface | Surface | Surface |
| Downwelling Longwave Radiation | W m$^{-2}$ | Surface | Surface | Surface |


**2.3      Data processing and bias correction workflow**
The workflow for the multi-stage bias correction of WFDEI against GEM-CaPA is shown in **Fig. 3**.
Bias correction was done after aggregating 1-h GEM-CaPA estimates to 3-h (the values at each time step
represent the mean of the previous 3-h period, to make it consistent with WFDEI) and interpolating both
WFDEI and GEM-CaPA to 0.125° resolution. For bias correction, a multi-stage approach was implemented
as follows. A multivariate generalization of the quantile mapping technique (MBCn, Cannon, 2018) which
combines quantile delta mapping (Cannon et al., 2015) and random orthogonal rotations to match the
multivariate distributions of two datasets was implemented to bias-correct WFDEI against GEM-CaPA at
3-h*0.125° resolution during the 2005-2016 period. The rationale for selecting the above bias correction
method is based on fitness for purpose, i.e. the method accounts for dependence between variables and
corrects multiple measures of joint dependence — attributes that can be important for hydrological
simulations (Meyer et al., 2019) — to preserve the physical realism of the corrected climate as much as
possible. Models were fitted to data for each calendar month while accounting for inter-variable
dependence structure. Using the fitted models (2005-2016), a hindcast was made of WFDEI between
1979-2004. Finally, the corrected WFDEI data derived from the fitted (2005-2016) and hindcast (1979-
2004) periods were concatenated to obtain the bias-corrected WFDEI-GEM-CaPA product (1979-2016).

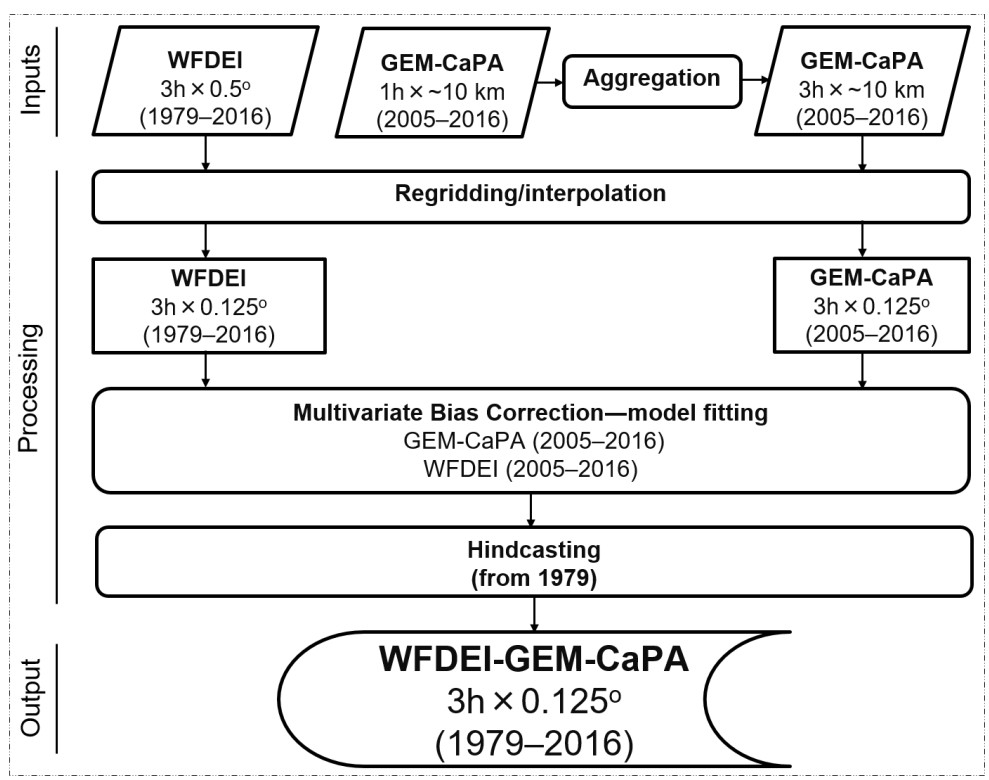

**Fig. 3**: A schematic representation of inputs and bias correction procedure used
to produce the WFDEI-GEM-CaPA meteorological forcing dataset
For bias-correcting the 15-member CanRCM4 initial condition ensemble against the WFDEI-GEM-
CaPA product, CanRCM4 was also spatially interpolated to match the WFDEI-GEM-CaPA specifications
using nearest neighbor interpolation. The multivariate bias correction technique (described above)
transfers all aspects of the WFDEI-GEM-CaPA continuous multivariate distribution to the corresponding
multivariate distribution of variables from CanRCM4 during the 1979 – 2008 calibration period (also used
here as historical period). Subsequently, when applied to future projections, changes in quantiles of each
variable between the historical and future period are also preserved. Models were fitted to data for each
calendar month and for each grid point while preserving the dependence structure among variables. The
historical datasets used in the fitting procedure include WFDEI-GEM-CaPA (1979 – 2008) and CanRCM4
(1979 – 2008). Using the fitted models, quantiles of CanRCM4 output from 1950 – 2100 were changed. To
evaluate the need to bias-correct CanRCM4, performance of the bias correction scheme, as well the
impact of bias correction on the climate change signal, the seasonal cycle of all 7 variables is assessed over
three 30-year periods: 1979–2008 (referred hereafter as 1990s); 2021–2050 (referred to hereafter as
2030s) and 2071–2100 (referred to hereafter as 2080s).
**3      Results and discussion**
**3.1     Bias correction of WFDEI**
Table 2 presents an overview of the seven variables processed in this study. Note that three of
the GEM variables (temperature, specific humidity, and wind speed) are at 40 m and are used directly to
correct the corresponding WFDEI surface variables (see **Table 2**). Therefore, the corrected WFDEI-GEM-
CaPA data for those 3 variables reflect 40 m elevations above the surface. The spatial coverage of the
WFDEI-GEM-CaPA data is the same as the areal extent of the MRB (Figs. 1 and 2). The suitability of the
bias correction algorithm to reproduce the observed seasonal cycle and inter-annual variability of the
variables was assessed for the fitting (2005-2016) and hindcast (1979-2004) periods. Data extracted over
the entire Mackenzie River basin is used to demonstrate the quality of the bias correction exercise and
uniqueness of the resulting output. **Fig. 4** shows the seasonal cycle for GEM-CaPA, WFDEI and WFDEI-
GEM-CaPA during the fitting period. Overall, the monthly distributions show that the bias was removed
for all variables resulting in the very close distributions between GEM-CaPA and WFDEI-GEM-CaPA. The
bias was particularly large for wind speed, an important variable for both alpine and prairie blowing snow
redistribution calculations (Pomeroy and Li, 2000), but was successfully removed. **Fig. 5** shows the mean
annual time series of the seven variables over the 1979-2016 period. It is noticeable that the bias is
corrected while the inter-annual variability is well preserved between WFDEI and WFDEI-GEM-CAPA,
except for shortwave radiation where the inter-annual variability is not fully preserved as shown by the
correlation between the WFDEI and WFDEI-GEM-CaPA annual series. However, this should not be a major
issue when impact models are driven using these data.The foregoing analyses have shown that the bias
in the WFDEI data was removed for both the fitting and hindcast periods. However, some potential
limitations remain—for example, WFDEI was interpolated directly from 0.5° to 0.125° and bias-corrected
against GEM-CaPA at 0.125°. The interpolation does not add any event-scale spatial variability for a
variable like precipitation which is very variable across different scales. These issues have been reviewed
extensively by Cannon (2018), Maraun (2013), Maraun et al. (2010), and Storch (1999).

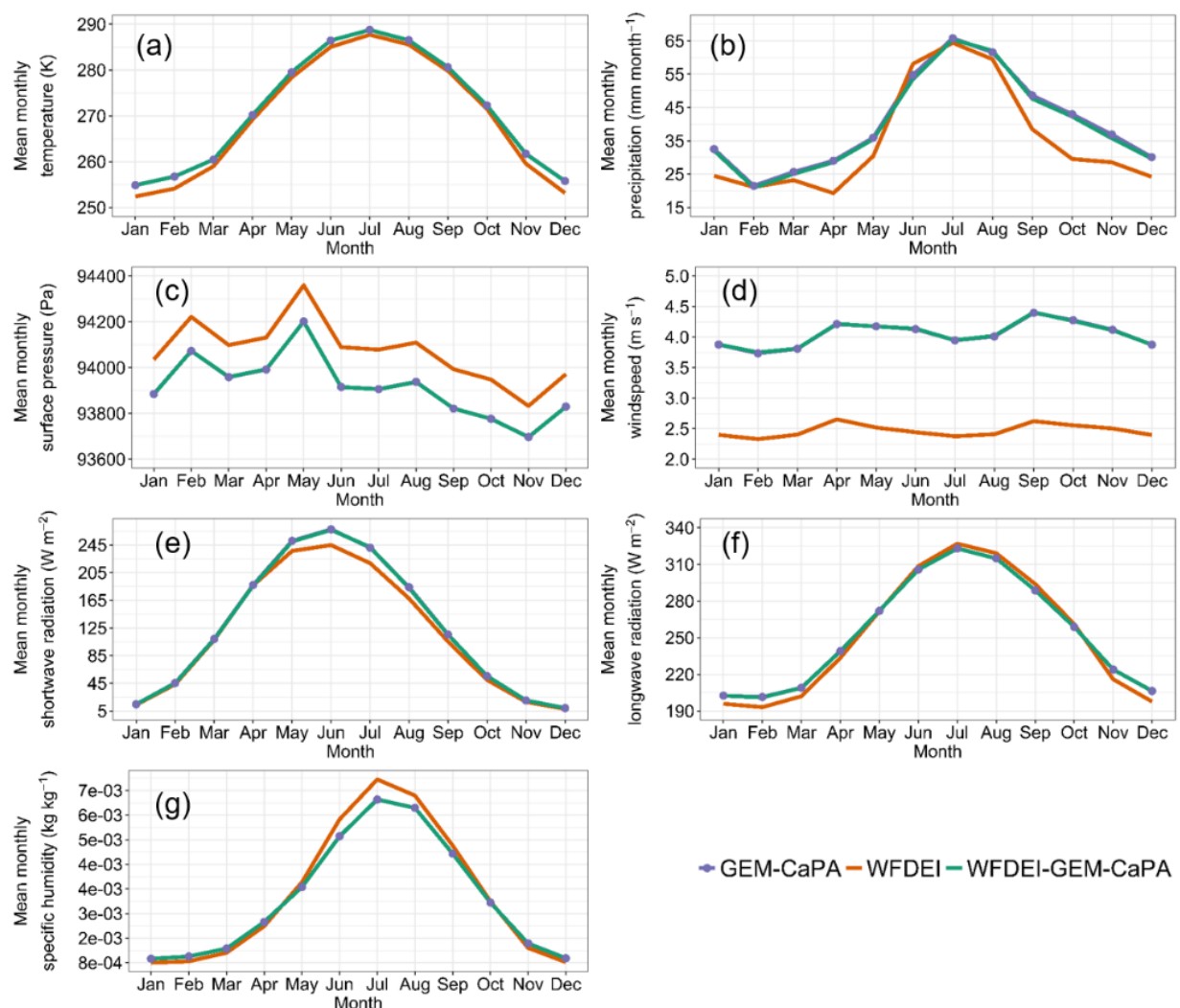

**Fig. 4**: Seasonal cycle of GEM-CaPA (dark slate blue), WFDEI (orange) and bias corrected data—WFDEI-
GEM-CaPA (green) for air temperature (a), precipitation (b), surface pressure (c), wind speed (d),
shortwave radiation (e), longwave radiation (f), and specific humidity (g) during the fitting period (2005-
341 2016)


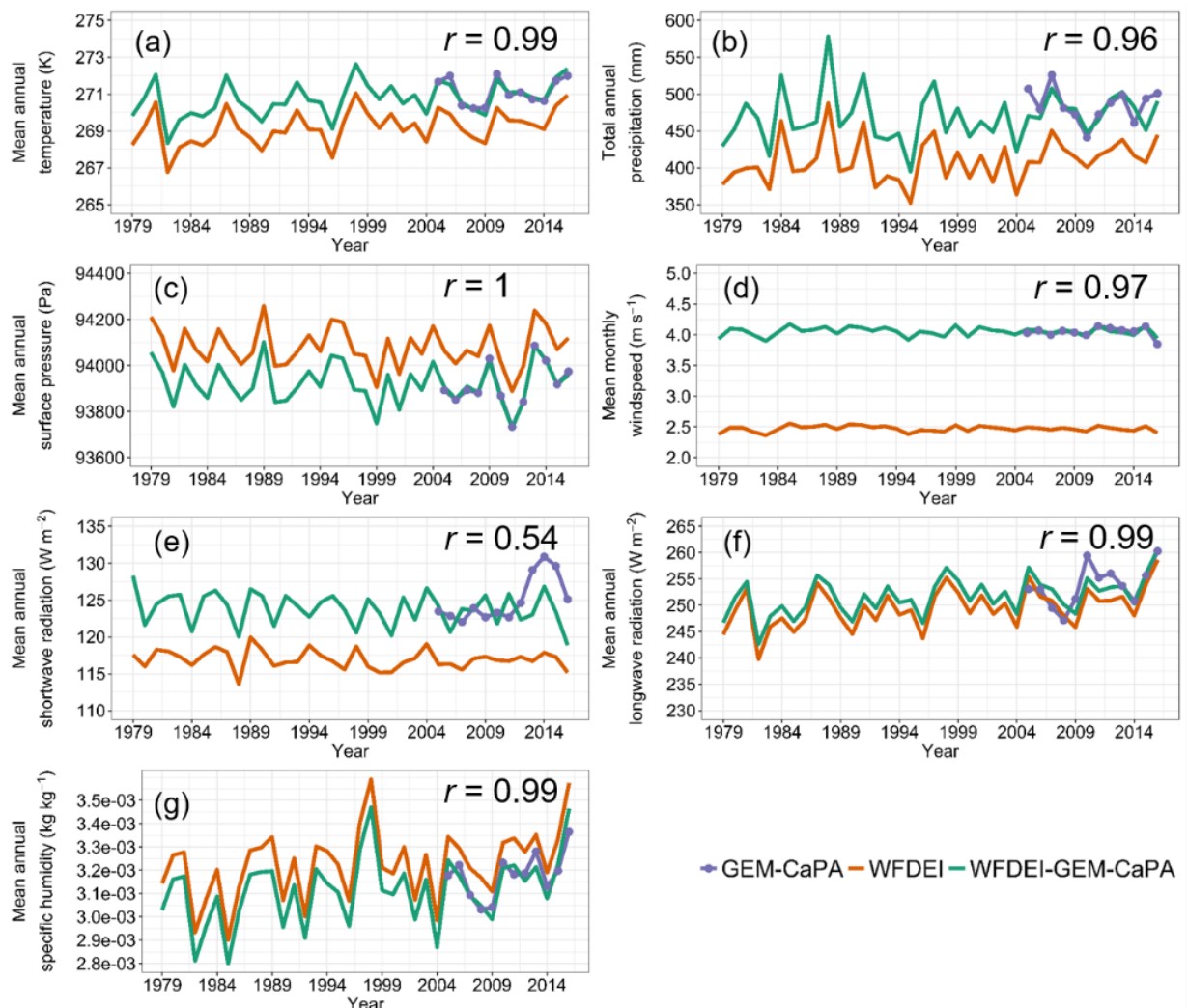

**Fig. 5**: Time series of GEM-CaPA (dark slate blue), WFDEI (orange) and bias corrected data—WFDEI-
GEM-CaPA (green) for air temperature (a), precipitation (b), surface pressure (c), wind speed (d),
shortwave radiation (e), longwave radiation (f), and specific humidity (g) during the periods 2005-2016
(GEM-CaPA) and 1979-2016 (WFDEI and WFDEI-GEM-CaPA). The correlation (r) between the WFDEI and
WFDEI-GEM-CaPA annual series is indicated for each variable.
**3.2    Validation of gridded products against station observations**

In this section, the WFDEI-GEM-CaPA product is validated against station observations (ECCC-S)

to indicate the benefit of bias-correcting WFDEI against GEM-CaPA. As mentioned in Section 2.2.3, the
validation focusses on variables for which station data could be found. Thus, shortwave and longwave
radiation are not validated as we could not find station data for those in ECCC-S data. The height
differences for temperature, humidity and wind speed between GEM-CAPA and WFDEI-GEM-CaPA (40 m)
on one side and ECCC-S data (surface) on the other introduce some inconsistencies that are discussed

below. Indirect validation is recommended for other variables through other means such as hydrological modelling. Validation is performed for the 2005 – 2016 period using daily totals for precipitation and daily averages for other variables. To compare stations against gridded products, the corresponding time series of gridded products for each gauge were obtained from the cell that contained the gauge (i.e. nearest neighbor) and were aggregated to the daily time scale.

**Fig. 6** depicts quantile–quantile (Q–Q) plots of daily precipitation from WFDEI-GEM-CaPA, WFDEI and GEM-CaPA compared against ECCC-S. As expected, although with noticeable differences across the MRB, CaPA agrees better with ECCC-S than WFDEI since some or all of these meteorological stations are assimilated by the CaPA system. Large daily amounts are generally underestimated by CaPA but CaPA sometimes overestimates these as well (e.g. Uranium City (AUT) station). WFDEI tends to underestimate the observed precipitation amounts at most stations except at Jasper Warden where it slightly overestimates small and moderate amounts. Bias correction brings WFDEI closer to CaPA for most stations but some biases remain, especially at the high ends of the distributions.

**Fig. 7** shows quantile–quantile (Q–Q) plots of mean daily temperature for the three gridded datasets versus ECCC-S. WFDEI is performing generally well for temperature except for low temperatures at Inuvik (the most northerly station). Despite the height difference (see Section 2.2.3), GEM is also close to observations for most stations with some overestimation of low temperatures. The temperature differences between the surface and the 40m level are generally small (1-2°C) at the daily scale. Given that temperature biases in WFDEI were small, WFDEI-GEM-CaPA is almost identical to GEM, i.e. all biases are removed.

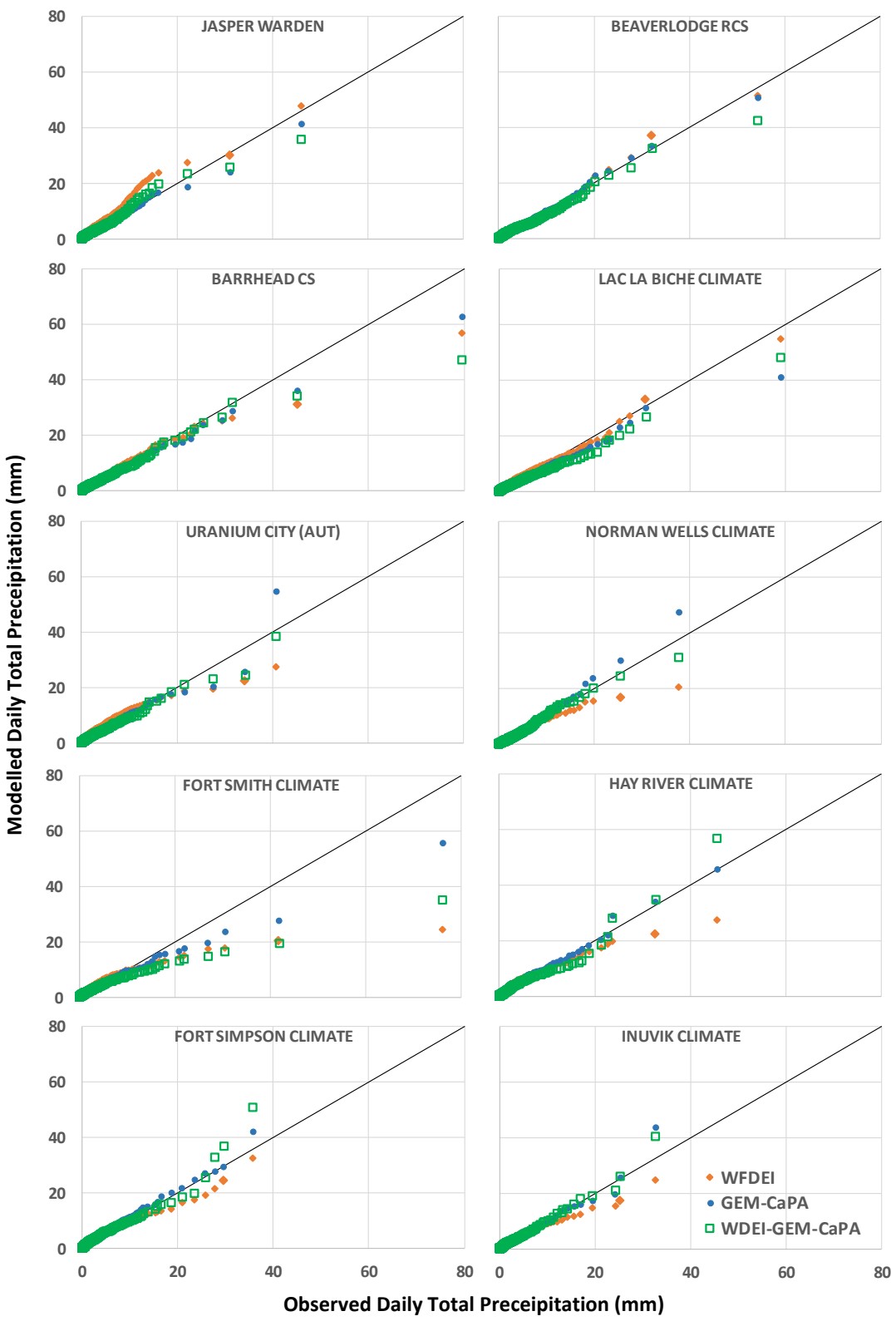

**Fig. 6**: Quantile-quantile plots of modelled (GEM-CaPA, WFDEI and WFDEI-GEM-CaPA) and observed (ECCC-S) daily total precipitation

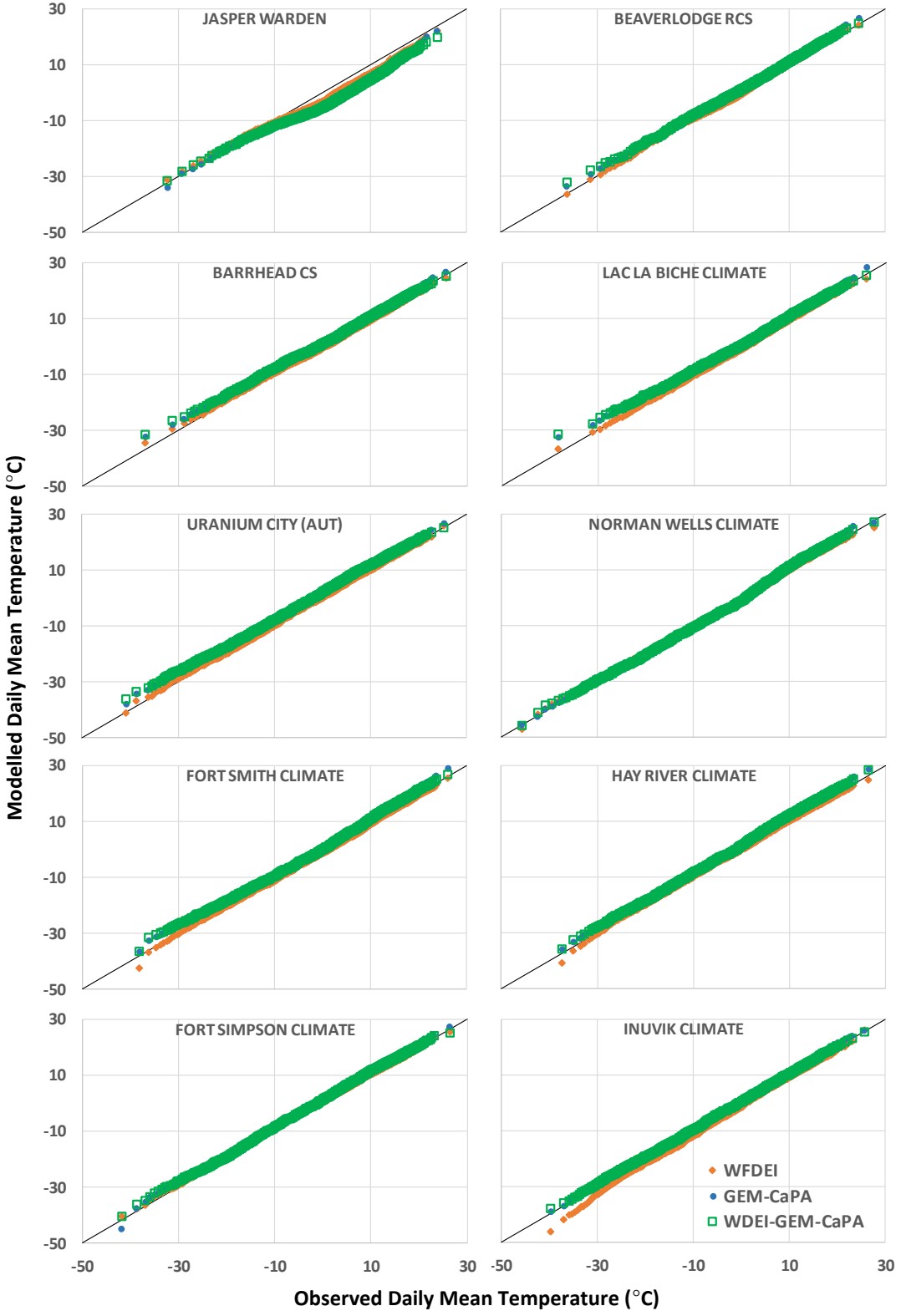

**Fig. 7**: Quantile-quantile plots of modelled (GEM-CaPA, WFDEI and WFDEI-GEM-CaPA) and observed (ECCC-S) daily mean temperature

Comparisons between gridded datasets and stations for daily mean surface pressure, wind speed,

and humidity are shown in Figs. 8, 9, and 10 respectively. WFDEI is generally performing well for surface
pressure (**Fig. 8**) such that bias correction seems unnecessary at most locations. Both datasets (WFDEI and
GEM) underestimate surface pressure at Jasper Warden station, which is at a relatively high elevation.
GEM is worse than WFDEI for this station and thus bias correction against GEM-CaPA deteriorates the
results. WFDEI slightly underestimates surface pressure at Uranium City (AUT) and Norman Wells Climate
stations but because GEM is close to observations, bias correction makes WFDEI-GEM-CaPA close to
observations at those two stations.

Mean daily wind speed (**Fig. 9**) is underestimated by WFDEI for most stations, especially at high

speeds. GEM winds are generally higher (except for Fort Simpson) because of the higher elevation (40 m)
of the dataset and thus the comparison to ECCC-S data is not favorable for this variable. It is generally
expected that wind speed increases with height. Bias correction of WFDEI against GEM-CaPA removes
differences between the two datasets and the resultant wind speed, thus, reflects the higher speeds to
be expected at 40 m.

Both WFDEI and GEM are close in terms of specific humidity at most stations (**Fig. 10**) despite the

height difference, with few exceptions. For example, humidity at Jasper Warden, Barrhead CS and Inuvik
Climate is underestimated by both WFDEI and GEM, especially at high values. Bias correction brings WFDEI
closer to GEM and thus results in improvements only if GEM is closer to observations than WFDEI. Thus,
results at Fort Smith Climate and Inuvik Climate stations are worse with bias correction. However, the bias
correction does not change the quantiles by much for most stations.

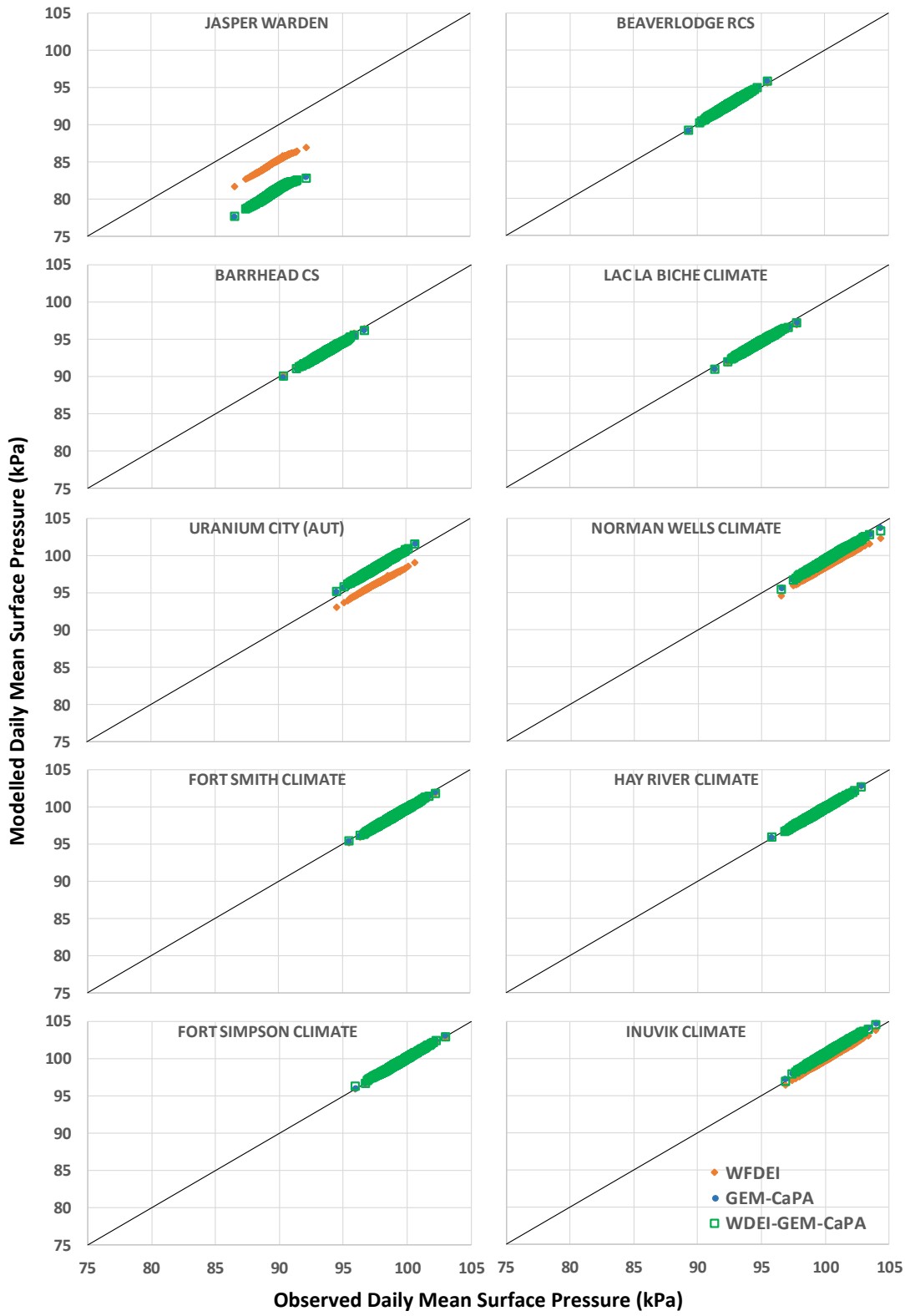

**Fig. 8**: Quantile-quantile plots of modelled (GEM-CaPA, WFDEI and WFDEI-GEM-CaPA) and observed daily mean surface pressure

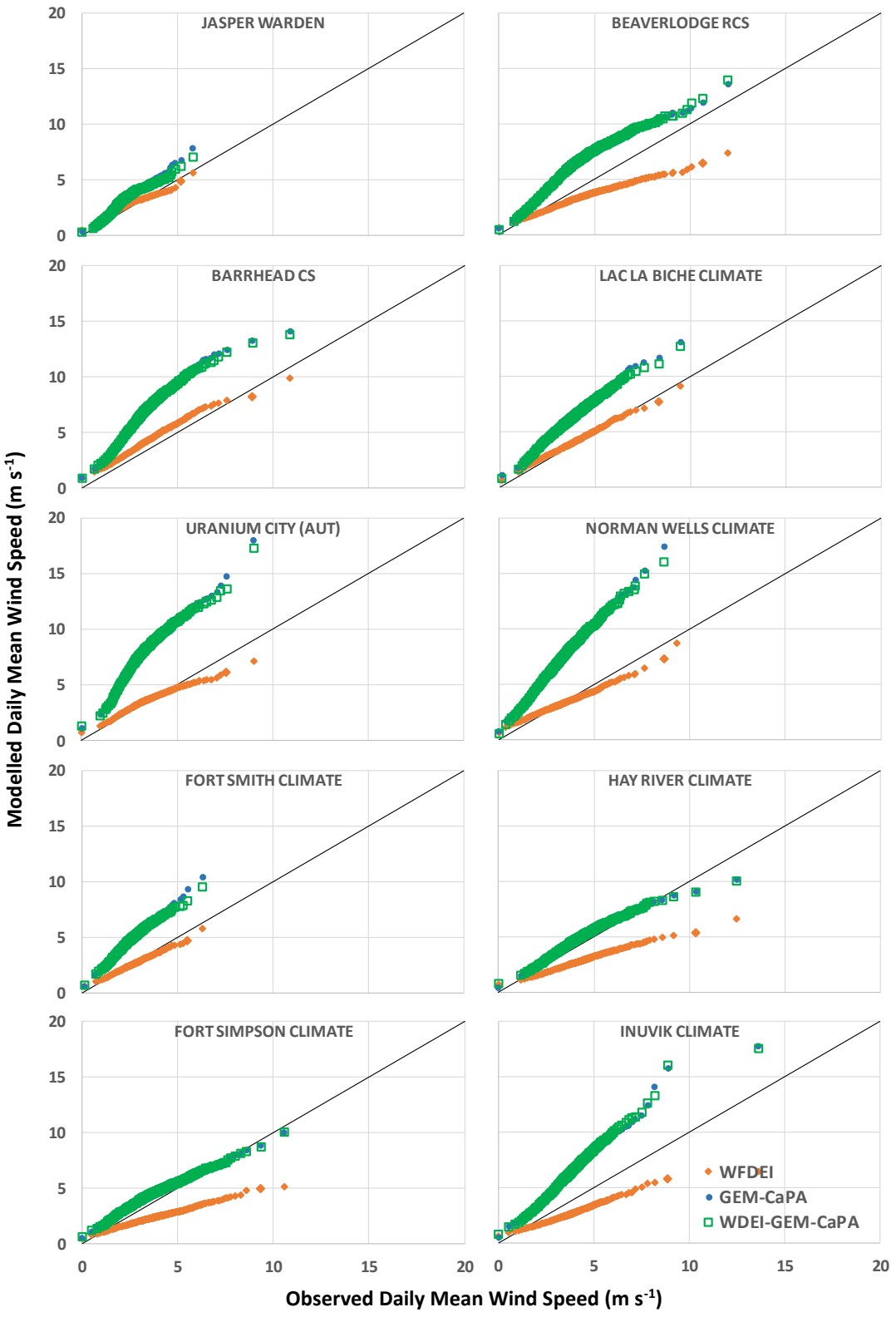

**Fig. 9**: Quantile-quantile plots of modelled (GEM-CaPA, WFDEI and WFDEI-GEM-CaPA) and observed daily mean wind speed

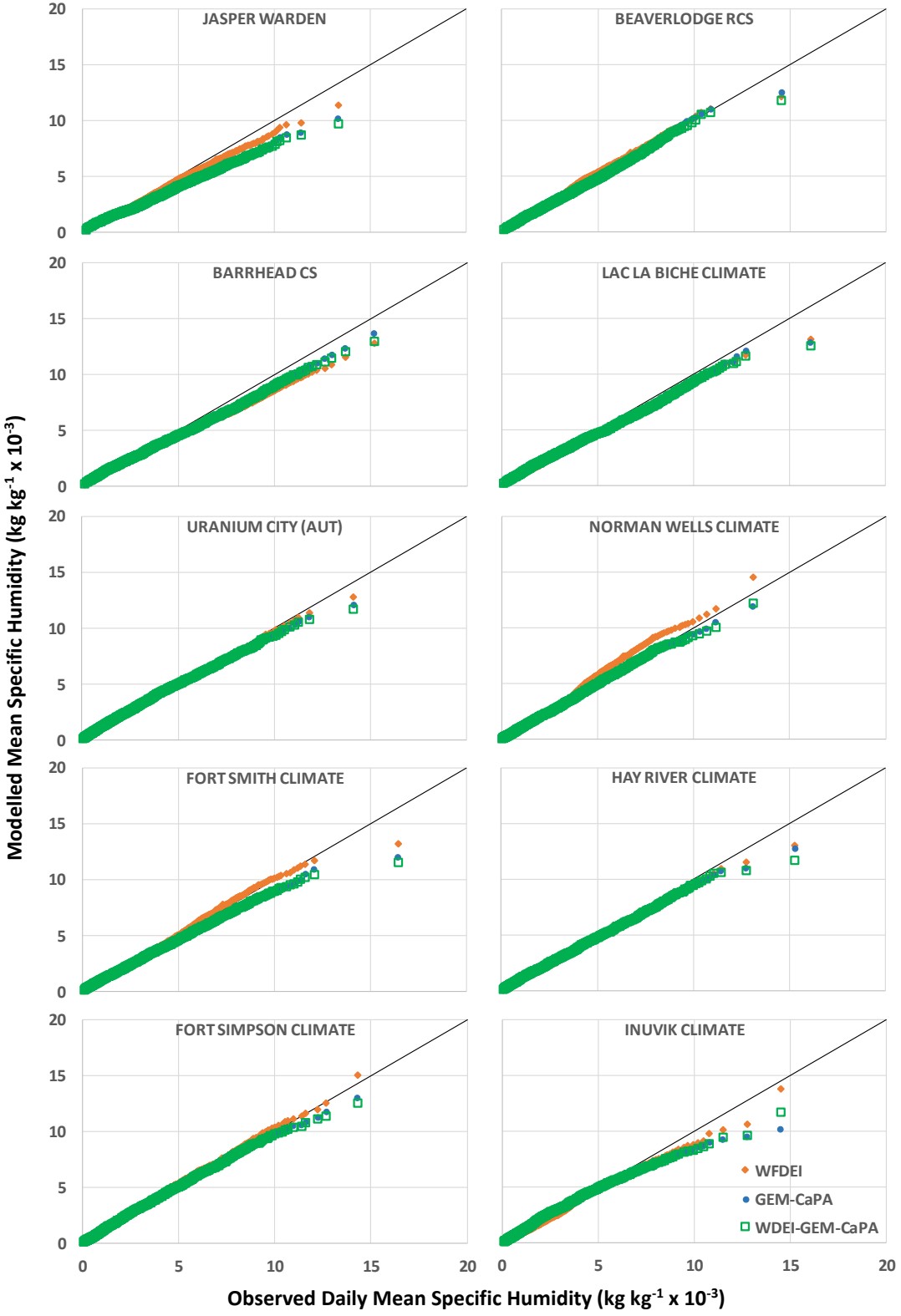

**Fig. 10**: Quantile-quantile plots of modelled (GEM-CaPA, WFDEI and WFDEI-GEM-CaPA) and observed daily specific humidity

Overall, GEM-CaPA performs similar to or better than WFDEI for most variables at the studied
stations, especially precipitation. Therefore, correcting WFDEI against GEM-CaPA adds value to the WFDEI
dataset and leads to a closer agreement between WFDEI-GEM-CaPA and ECCC-S. Precipitation is one of
the most important variables and most difficult to correct. Note that extracting data from grid points does
not only have the effect of smoothing the areal averages but comparing grid point estimates to station
values may not provide a clear picture of the quality of a gridded product. However, this diagnostic
analysis provides preliminary insights into the potential performance of a dataset.
**3.2     Bias correction of future climate projections**
In this section, the need to bias-correct the CanRCM4 outputs is shown and whether the simulated
climate change signal was preserved after applying MBCn to the CanRCM4 outputs is determined. **Fig. 11**
shows the climatological seasonal cycle of all 7 variables which are required to drive the MESH model for
the MRB. First, between April and October, CanRCM4 overestimates the observed (i.e. WFDEI-GEM-CaPA)
daily precipitation amounts and specific humidity during the historical period.  This is also true in the case
of daily mean wind speed in the cold months (October to April). However, it underestimates the wind
speed in the warm season (May to September). Surface pressure is underestimated during September to
May and overestimated in the summer (June to August). For the other variables (e.g. air temperature and
radiation), CanRCM4 can simulate the observed seasonal cycle closely although biases still exist. These
biases necessitated the application of the MBCn algorithm on the raw CanRCM4 outputs. The MBCn
algorithm removed the bias in the CanRCM4 simulations during the fitting period (1990s) as can be judged
from the close fit between WFDEI-GEM-CaPA and the unbiased CanRCM4 output (corr_1990s). On the
projected climate change signal, there is a projected change in the amplitude of all variables but not a
shift in the phase of the cycle over the MRB with global warming. Precipitation, specific humidity and
longwave radiation are projected to increase in the future, with larger changes expected in the warm
season (April – October) while air temperature is projected to increase, particularly in the cold months
(October – March). These climate change signals are very well preserved after applying MBCn to the
CanRCM4 simulations.

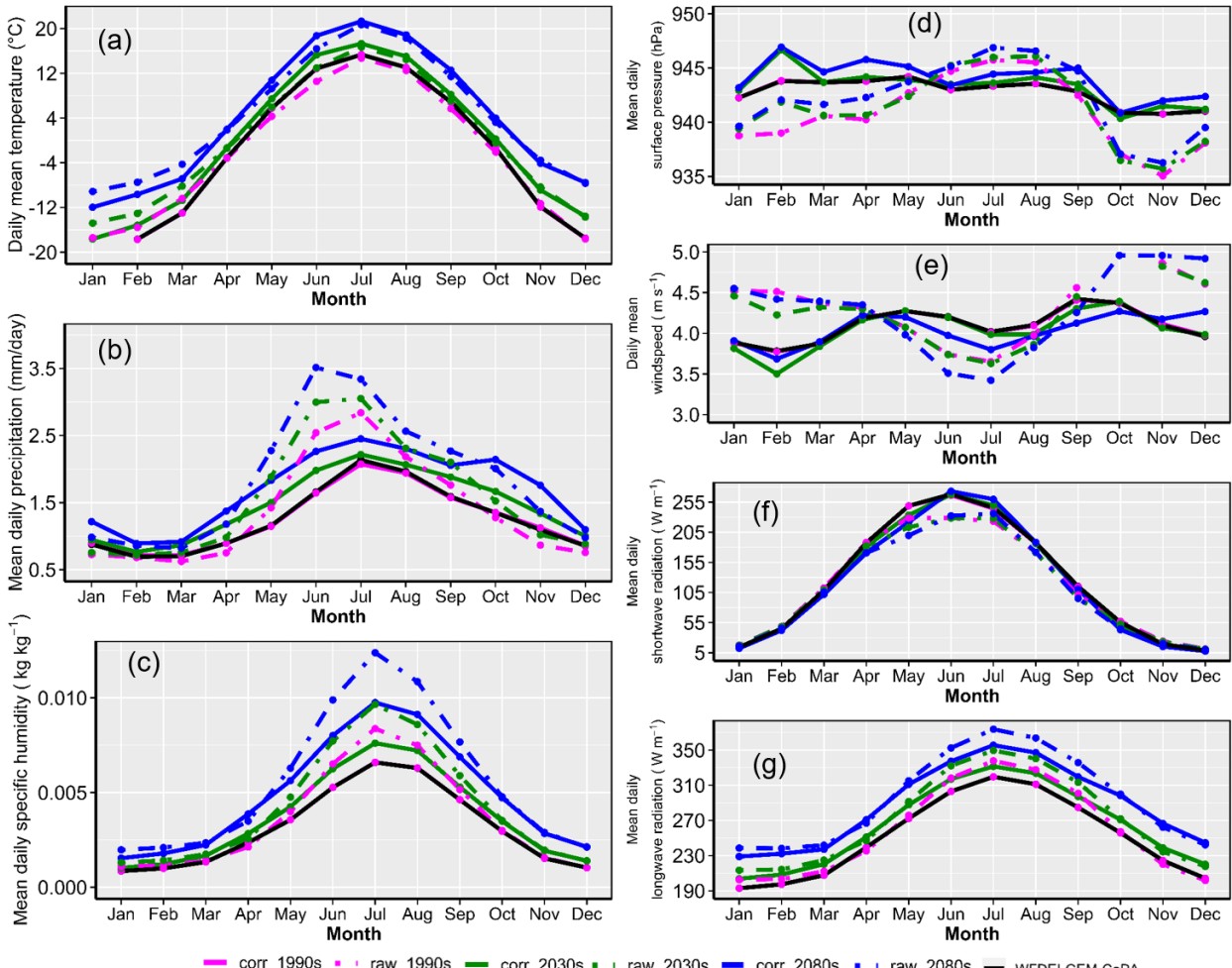

**Fig. 11**: Seasonal cycle of WFDEI-GEM-CaPA, raw and bias-corrected CanRCM4 data for air temperature
(a), precipitation (b), specific humidity (c), surface pressure (d), wind speed (e), shortwave radiation (f),
and longwave radiation (g) during the periods 1979–2008; 2021–2050 and 2071–2100.
**4      Conclusions**

Cold regions hydrology is very sensitive to the impacts of climate warming. More physically

realistic hydrological models need to be driven by reliable climate forcing and can provide the capability
to assess hydrological responses to climate variability and change. However, cold regions such as the
Mackenzie River Basin often have sparse surface observations, particularly at high elevations and latitudes
where a large amount of runoff is generated, or important cyrspheric processes are impacting the
hydrology   A novel approach to developing a long-term data set using the  WFDEI-GEM-CaPA approach
outlined above was necessary to better understand and represent the seasonal/inter-annual variability of
hydrological fluxes and the timing of runoff, and their long-term trends. This dataset is also valuable for
bias correction of climate model projections to assess potential impacts of future climate change on the
hydrology and water resources of the basin.

The raw CanRCM4 outputs were found to have systematic biases, which required bias correction

towards WFDEI-GEM-CaPA. There are clear discrepancies between the seasonal cycle of WFDEI-GEM-
CaPA, raw, and bias-corrected CanRCM4 data. For example, the CanRCM4 simulated climatological daily
mean precipitation in June over the MRB between 1979 – 2008 is ~2.5 mm/day while the observed value
is ~1.5 mm/day. This results in a 1.0 mm/day wet bias which can have various implications for quantifying
water resources availability, management and adaptation in a future changed climate. Therefore, it was
crucial to produce the bias-corrected CanRCM4 outputs prior to using the data to drive large scale
hydrological models for climate change impacts analysis in the MRB. Nevertheless, the WFDEI-GEM-CaPA
dataset, used here as the reference, has uncertainties (although it is superior to WFDEI as shown in Figs.
6-11) and should be used with caution especially from the perspective of over-interpreting impact model
outputs.
**Data availability**

The final product (WFDEI-GEM-CaPA, 1979-2016) is freely available at the Federated Research

Data Repository at http://dx.doi.org/10.20383/101.0111 (Asong et al., 2018) while the original (raw) and
corrected CanRCM4 data are also freely available at https://doi.org/10.20383/101.0162 (Asong et al.,

2019).

**Author contribution**

Z.E., H.W., J.P., A.P., and M.E. conceived and designed the experiment. D.P. preprocessed the

GEM-CaPA data, A.C. developed the bias correction model code and guided the computing procedures
while Z.E. performed the computations. M.E extracted the sample data used in generating Figures 4 and
5. M.E. and Z.E. performed the validation against station observations. Z.E. and M.E. prepared the
manuscript with contributions from all co-authors.
**Competing interests**

The authors declare that they have no conflict of interest.

**Acknowledgements**

Financial support from the Canada Excellence Research Chair in Water Security, the NSERC

Changing Cold Regions Network and the Global Water Futures program is gratefully acknowledged.
Thanks are due to the Meteorological Service of Canada for providing access to the GEM-CaPA data used
in this study. We also thank Dr. Graham Weedon for making available the WFDEI dataset. We also
appreciate the efforts of Amber Peterson, Data Manager, Global Institute for Water Security toward
archiving the data at the Federated Research Data Repository.

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
