# Peer review of "High-Resolution Meteorological Forcing Data for Hydrological Modelling and Climate Change Impact Analysis in Mackenzie River Basin"

_Earth System Science Data, 2019_

## Referee Comment (RC1) · Anonymous Referee #1 · 16 Aug 2019

This paper describes the procedures used to generate two bias-corrected data products WFDEI-GEM-CaPA and CanRCM4 for hydrological modelling and climate change impact analysis in Mackenzie River basin. The uniqueness of WFDEI-GEM-CaPA data is from combining the WFDEI and GEM-CaPA data sets with multivariate bias correction and model fitting methods. WFDEI data set has a longer record period (1979-2016), and is biased from the observations in Mackenzie River basin. The GEM-CaPA data set has a shorter record period (2005-2016), and matches well with the observations in Mackenzie River basin. As a result of the combining, the record length of the combined WFDEI-GEM-CaPA data is the same with that of WFDEI and the biases in WFDEI data were corrected. The bias-corrected CanRCM4 data product was ob-

[Figure]

tained from the modified CanRCM4 output (1950-2100) after using the WFDEI-GEM-CaPA data product to correct the 15-member CanRCM4 initial condition ensemble in the historical period (1979-2008). The data products are of significance in developing distributed hydrological models and improving the assessment of the climate change impacts over the Mackenzie River basin.

Overall, the paper was written and organized very well. It introduces the data sources and explains how these data sources were selected and used. It clearly presents the outline of the multi stages of data processing and bias corrections. It uses station observations to validate the bias corrections and the provided results prove the added values of the two data products as a constant set of historical and climate projection data for large-scale modeling and future climate scenario analysis.

Specifically, the paper does not give any details about how the multivariate generalization of the quantile mapping technique was applied to the data sources and how the model fitting was done. More detailed explanations with examples would help readers who are not familiar with this mapping method. A brief discussion about this method's advantages compared to other methods is also helpful. The authors may need to explain clearly why the air temperature, specific humidity and wind speed of GEM-CaPA at 40-m height can be directly used to correct the WFDEI biases of these variables, but the station surface level observations of these variables cannot be inversely used for the validation of bias corrections at 40-m height. Because the surface pressure provided by all data sources is at surface level, the station observations of surface pressure should also be used to validate the bias corrections in addition to the precipitation, though the surface pressure as a forcing variable may not be as important as the precipitation in hydrological models.

A couple minor things include: 1) Line 243 on page 12, to keep the consistency throughout the paper, suggest to replace "0.44 degrees" with the same format as in line 235 on the same page. 2) Suggest to enlarge a bit Figure 4 and Figure 5 to improve their legibility.

---

## Referee Comment (RC2) · Anonymous Referee #2 · 24 Aug 2019

The data description paper entitled "High-Resolution Meteorological Forcing Data for Hydrological Modelling and Climate Change Impact Analysis in Mackenzie River Basin" describes a new gridded climate reanalysis data set (WFDEI-GEM-CaPA) and its application to correct biases in CanRCM4 data set over the Mackenzie River Basin. The data set consist of seven hydro-climatic variables that are required to run a distributed and process based hydrologic model over the basin. The original data sets used to generate the new gridded data are described briefly and the methodology followed in blending the original data as well as validating the resulting new data set is well described. The resulting bias corrected CanRCM4 data is also shown to preserve the climate projection signals while removing the biases on monthly basis. In general, the

manuscript describes very well the steps in preparing the new data sets as well as the added value gained from these new data sets.

However, I have also made the following few observations that may require some explanation to further improve the quality of the manuscript:

1. Page 14; Lines 270 – 272: I do not understand why the historical period for bias correcting the CanRCM4 data is chosen to be 1979 – 2008 instead of 1979 -2005. We all know that the CMIP5 standard is to apply the historical emission rate until 2005 and then use the various emissions scenarios from 2006 onward. That means the 2006 – 2008 period is a climate projections period, not historical climate period. I would like to ask the authors to provide a convincing argument for this discrepancy.

2. Page 14; Lines 278 – 281: the authors wrote ". . . the GEM 40 m variables are used directly to correct WFDEI surface level variables (2 m temperature, 2 m specific humidity, and 10 m wind speed). Therefore, the corrected WFDEI-GEM-CaPA data reflect 40 m elevations above the surface." These two statements seems to be contradictory and do not make sense to me at all. Other paragraphs (including Table 2) seems to suggest that the GEM 40 m variables are used to correct the WFDEI 40 m variables; not the WFDEI surface variables. The authors should correct or explain these discrepancies.

3. Page 17; Lines 318 – 319: the authors wrote ". . . the height differences preclude direct validation of other variables against the ECCC-S data which are measured at the surface." However, Table 2 shows at least other three surface variables (pressure, short and long wave radiation) that can be used for direct validation. Therefore, the authors have to explain why those additional surface variables were not used for validation.

4. Page 17; Lines 325 – 327: To compare station precipitation values against gridded products, the authors chose to interpolate the surrounding four grid cells. Knowing that averaging gridded products has under estimation (smoothing) effect; why didn't they choose to use only the closest grid to the station data for better comparison?

5. While the gridded data set produced and explained in this paper is aimed to be used by the MESH model that is capable of using climatic forcing data at 40 m above surface level, most other process based hydrologic models need forcing data near the surface (such as 2m for temperature and 10 m for wind). So, why not also produce the corresponding surface level WFDEI-GEM-CaPA and bias corrected CanRCM4 values for those variables.

---

## Author Comment (AC1) · 20 Jan 2020

The comment was uploaded in the form of a supplement:
https://www.earth-syst-sci-data-discuss.net/essd-2019-103/essd-2019-103-AC1-supplement.pdf

---

## Author Comment (AC2) · 20 Jan 2020

We thank the reviewer for reviewing our manuscript and providing useful comments which helped improve the quality of the analysis and structure of the paper. Please find below replies to comments. Comments are listed first (in black) followed by replies (in blue). Changes in the manuscript are highlighted in green colour.

This paper describes the procedures used to generate two bias-corrected data products WFDEI-GEM-CaPA and CanRCM4 for hydrological modelling and climate change impact analysis in Mackenzie River basin. The uniqueness of WFDEI-GEM-CaPA data is from combining the WFDEI and GEM-CaPA data sets with multivariate bias correction and model fitting methods. WFDEI data set has a longer record period (1979-2016) and is biased from the observations in Mackenzie River basin. The GEM-CaPA data set has a shorter record period (2005-2016) and matches well with the observations in Mackenzie River basin. As a result of the combining, the record length of the combined WFDEI-GEM-CaPA data is the same with that of WFDEI and the biases in WFDEI data were corrected. The bias-corrected CanRCM4 data product was obtained from the modified CanRCM4 output (1950-2100) after using the WFDEI-GEM-CaPA data product to correct the 15-member CanRCM4 initial condition ensemble in the historical period (1979-2008). The data products are of significance in developing distributed hydrological models and improving the assessment of the climate change impacts over the Mackenzie River basin. Overall, the paper was written and organized very well. It introduces the data sources and explains how these data sources were selected and used. It clearly presents the outline of the multi stages of data processing and bias corrections. It uses station observations to validate the bias corrections and the provided results prove the added values of the two data products as a constant set of historical and climate projection data for large-scale modeling and future climate scenario analysis.

Thank you.

Specifically, the paper does not give any details about how the multivariate generalization of the quantile mapping technique was applied to the data sources and how the model fitting was done. More detailed explanations with examples would help readers who are not familiar with this mapping method. A brief discussion about this method's advantages compared to other methods is also helpful.

We thank the reviewer for requesting further details on the modelling methodology. To address the reviewer's concern, in Section 2 (Methodology) sub-section 2.3, emphasis is placed on the data processing and bias correction workflow (Figure 3) wherein details on each step of model fitting, input and output data are described.

As the focus of the analysis is not on the mapping method itself but its application, Cannon (2018) and Cannon et al. (2015) have been referenced for details of the methodology applied in this study. As for the rationale for selecting this bias correction method, the following sentence has been added to Section 2.3:

'The rationale for selecting the above bias correction method is based on its suitability for our purpose, that is, the method accounts for dependence among variables and corrects multiple measures of joint dependence to preserve the physical realism of the corrected climate as much as possible'.

The general advantages of the selected method compared to others are discussed in the detail in Cannon (2018).

The authors may need to explain clearly why the air temperature, specific humidity and wind speed of GEM-CaPA at 40-m height can be directly used to correct the WFDEI biases of these variables, but the station surface level observations of these variables cannot be inversely used for the validation of bias corrections at 40-m height. Because the surface pressure provided by all data sources is at surface level, the station observations of surface pressure should also be used to validate the bias corrections in addition to the precipitation, though the surface pressure as a forcing variable may not be as important as the precipitation in hydrological models.

We have re-considered all variables for which we could find data in the ECCC archive and thus we rewrote Sections 2.2.3 and 3.2 to present validation results for temperature, surface pressure, humidity and wind speed while noting that the height difference can introduce biases in the validation. We could not find radiation data for validation.

A couple minor things include:

1) Line 243 on page 12, to keep the consistency throughout the paper, suggest to replace "0.44 degrees" with the same format as in line 235 on the same page.

Done! Thank you.

2) Suggest to enlarge a bit Figure 4 and Figure 5 to improve their legibility.

We made multiple attempts to enlarge the two figures through sensitivity analysis of different font sizes. However, the current font size is that which ensures text in the Figures is printed when the Figures are exported. We have a very high-resolution image for each Figure which would be made available to the journal during manuscript production. The image resolution reduces when inserted into word.

We thank the reviewer for reviewing our manuscript and providing useful comments which helped improve the quality of the analysis and structure of the paper. Please find below replies to comments. Comments are listed first (in black) followed by replies (in blue). Changes in the manuscript are highlighted in green colour.

The data description paper entitled "High-Resolution Meteorological Forcing Data for Hydrological Modelling and Climate Change Impact Analysis in Mackenzie River Basin" describes a new gridded climate reanalysis data set (WFDEI-GEM-CaPA) and its application to correct biases in CanRCM4 data set over the Mackenzie River Basin. The data set consist of seven hydro-climatic variables that are required to run a distributed and process based hydrologic model over the basin. The original data sets used to generate the new gridded data are described briefly and the methodology followed in blending the original data as well as validating the resulting new data set is well described. The resulting bias corrected CanRCM4 data is also shown to preserve the climate projection signals while removing the biases on monthly basis. In general, the manuscript describes very well the steps in preparing the new data sets as well as the added value gained from these new data sets.

Thank you for summarizing our work and acknowledging the importance and challenges that come with data fusion in these sparsely gauged cold regions.

However, I have also made the following few observations that may require some explanation to further improve the quality of the manuscript:

1. Page 14; Lines 270 – 272: I do not understand why the historical period for bias correcting the CanRCM4 data is chosen to be 1979 – 2008 instead of 1979 -2005. We all know that the CMIP5 standard is to apply the historical emission rate until 2005 and then use the various emissions scenarios from 2006 onward. That means the 2006 –2008 period is a climate projections period, not historical climate period. I would like to ask the authors to provide a convincing argument for this discrepancy.

The reviewer's concern about the 2006–2008 'climate projections' period is an important consideration which was discussed thoroughly internally before arriving at a final decision to include this period in the baseline window. First, Section 1 of the paper elaborates on the need to produce a longer record data set for hydrological modelling in the basin. GEM-CaPA (the best available product) is too short (2005-2016) to be used to correct CanRCM4 climate since the recommended minimum record length for bias correction of 'a climate' is 30 years. As WFDEI starts in 1979, using data for 1979-2005 will result in only 27 years of records—hence we added 2006–2008 to the 'observed' baseline to get 30 years data. From a logical perspective, the 2006–2008 data should not change significantly the main conclusions of the analysis. In terms of climate projections, the period 2006–2008 is still well within the initialization window of future emissions—thus we believe these data have close similarities with 1979-2005. Nonetheless, the data availability challenges necessitated the approach implemented in this study.

2. Page 14; Lines 278 – 281: the authors wrote ": : : the GEM 40 m variables are used directly to correct WFDEI surface level variables (2 m temperature, 2 m specific humidity, and 10 m wind speed). Therefore, the corrected WFDEI-GEM-CaPA data reflect 40 m elevations above the surface." These two statements seems to be contradictory and do not make sense to me at all. Other paragraphs (including Table 2) seems to suggest that the GEM 40 m variables are used to correct the WFDEI 40 m variables; not the WFDEI surface variables. The authors should correct or explain these discrepancies.

The phrase 'WFDEI surface level variables (2 m temperature, 2 m specific humidity, and 10 m wind speed)' is in fact confusing since 'surface level variables' such as precipitation and pressure are also included in the bias corrected variables as in Table 2. Unfortunately, there are no 40 m level variables available in WFDEI. That would have removed the concern about changing the height of observations altogether, if these were available. We clarified the statement as (Line 313): 'Note that three of the GEM variables (temperature, specific humidity, and wind speed) are at 40 m and are used directly to correct the corresponding WFDEI surface variables (see **Error! Reference source not found.**). Therefore, the corrected WFDEI-GEM-CaPA data for those 3 variables reflect 40 m elevations above the surface.'

3. Page 17; Lines 318 – 319: the authors wrote ": : : the height differences preclude direct validation of other variables against the ECCC-S data which are measured at the surface." However, Table 2 shows at least other three surface variables (pressure, short and long wave radiation) that can be used for direct validation. Therefore, the authors have to explain why those additional surface variables were not used for validation.

We have re-considered all variables for which we could find data in the ECCC archive and thus we rewrote Sections 2.2.3 and 3.2 to present validation results for temperature, surface pressure, humidity and wind speed while noting the height difference can introduce biases in the validation. We could not find radiation data for validation.

4. Page 17; Lines 325 – 327: To compare station precipitation values against gridded products, the authors chose to interpolate the surrounding four grid cells. Knowing that averaging gridded products has under estimation (smoothing) effect; why didn't they choose to use only the closest grid to the station data for better comparison?

We agree with the reviewer's concern which is a widely recognized challenge when comparing station data to gridded products. We have redone the validation results using nearest neighbor interpolation (i.e. the grid containing the gauge) as suggested.

5. While the gridded data set produced and explained in this paper is aimed to be used by the MESH model that is capable of using climatic forcing data at 40 m above surface level, most other process based hydrologic models need forcing data near the surface (such as 2m for temperature and 10 m for wind). So, why not also produce the corresponding surface level WFDEI-GEM-CaPA and bias corrected CanRCM4 values for those variables.

Thank for this suggestion. Surface data from GEM is only available for a very short period and thus we could not generate a surface WFDEI-GEM-CaPA dataset. However, we have corrected CanRCM4 surface data against WFDEI (despite its biases) and the data will be available soon at https://tuna.cs.uwaterloo.ca/.